# Super Resolution Generative Adversarial Network (SRGANs) for Wheat Stripe Rust Classification

**DOI:** 10.3390/s21237903

**Published:** 2021-11-26

**Authors:** Muhammad Hassan Maqsood, Rafia Mumtaz, Ihsan Ul Haq, Uferah Shafi, Syed Mohammad Hassan Zaidi, Maryam Hafeez

**Affiliations:** 1School of Electrical Engineering and Computer Science (SEECS), National University of Sciences and Technology (NUST), Islamabad 44000, Pakistan; mmaqsood.mscs19seecs@seecs.edu.pk (M.H.M.); ihaq.msee19seecs@seecs.edu.pk (I.U.H.); ushafi.dphd18seecs@seecs.edu.pk (U.S.); drzaidi@seecs.edu.pk (S.M.H.Z.); 2Department of Engineering and Technology, School of Computing and Engineering, University of Huddersfield, Queensgate, Huddersfield HD1 3DH, UK; m.hafeez@hud.ac.uk

**Keywords:** GANs, SRGANs, deep learning, super resolution, wheat stripe rust

## Abstract

Wheat yellow rust is a common agricultural disease that affects the crop every year across the world. The disease not only negatively impacts the quality of the yield but the quantity as well, which results in adverse impact on economy and food supply. It is highly desired to develop methods for fast and accurate detection of yellow rust in wheat crop; however, high-resolution images are not always available which hinders the ability of trained models in detection tasks. The approach presented in this study harnesses the power of super-resolution generative adversarial networks (SRGAN) for upsampling the images before using them to train deep learning models for the detection of wheat yellow rust. After preprocessing the data for noise removal, SRGANs are used for upsampling the images to increase their resolution which helps convolutional neural network (CNN) in learning high-quality features during training. This study empirically shows that SRGANs can be used effectively to improve the quality of images and produce significantly better results when compared with models trained using low-resolution images. This is evident from the results obtained on upsampled images, i.e., 83% of overall test accuracy, which are substantially better than the overall test accuracy achieved for low-resolution images, i.e., 75%. The proposed approach can be used in other real-world scenarios where images are of low resolution due to the unavailability of high-resolution camera in edge devices.

## 1. Introduction

Wheat is one of the most important crop in the world that is the main source of nutrients for around 40% of the global population. It also provides 20% of the daily protein and food calories to human beings. Economically, the importance of wheat is evident from the fact that the global trade for wheat is more than the global trade for all the other crops combined [1]. Owing to its critically important place in human life, there is a need for proactive approaches to combating the pests and diseases that affect the wheat crop on an annual basis. One of these diseases is yellow rust, which can have a devastating impact on the production of wheat. It can cause up to a 40% yield loss to a total harvest depending upon the severity of the disease.

Attempts to counter the impact of yellow rust on wheat include spraying the whole fields with fungicides which can be counterproductive and have adverse affects on healthy portions of the crop. Therefore, the early and accurate detection of the disease is highly desirable, and various approaches have been adopted for detecting wheat yellow rust in recent years including machine learning and deep learning methods [2,3,4,5]. However, all of these approaches have very specific needs for data in cases of machine learning and high-resolution and clear imagery in the cases of deep learning approaches.

The requirements such as clean and high-quality data for different approaches can be met in experimental and lab settings, but it can be very hard and expensive to meet these requirements in real-world conditions. This is especially true in the case of image data where not all commonly available devices are able to capture high-resolution images. Generative adversarial networks (GAN) [6] have been highly successful in removing various limitations imposed on data. These limitations include volume, quality and variety of the data. There has been an increased usage of GANs in different fields. The different variations of GANs are able to produce realistic images after training on similar sample data [6], while other variants of GANs, super-resolution GANs (SRGAN), have the ability to produce high-resolution images from low-resolution images [7].

Convolutional neural networks (CNN) have proven to be extremely potent on different learning tasks related to images. However, the performance of the CNNs is highly dependent on the quality of images used for training. High-resolution images result in better performance than low-resolution images. In real-world situations, where high-resolution image capturing devices are not readily available, this can become a challenge for improving the performance of CNNs. This challenge is also prevalent where high-resolution training data are available, but upon deployment, the image capturing devices are not able to capture data with similar resolutions, therefore, resulting in diminishing performances of the system. These issues are even more prominent in agricultural applications of CNNs where data acquisition is affected by different environmental factors.

In light of all these issues, this paper presents a system that tackles the challenge of the lack of availability of high-resolution images by harnessing the power of super-resolution GANs for wheat yellow rust detection. All the images used in this paper contain a single leaf roughly in the center of the image. This helps keep the focus of the paper on studying the feasibility of SRGANs for wheat yellow rust detection. The main contributions of this study are:End-to-end pipeline for wheat yellow rust detection in low-resolution imagesEmpirical analysis of effectiveness of SRGANs for wheat yellow rust detection

The rest of the paper is organized as follows: the related work is discussed in Section 2; the methodology is presented in Section 3; Section 4 presents the results and discussion and conclusion. In addition, future works are discussed in Section 5.

## 2. Related Work

Deep learning models and their performance have a direct correlation with the quality of the data used for training. High-quality data produce better results when compared with the results of low-quality data. This is especially true in cases where the data are composed of images. When the learning tasks are of critical importance, such as wheat yellow rust detection, the quality of the data becomes even more important as low-quality data can lead to faulty detection which can be counterproductive. This section discusses applications of GANs in general and different approaches that improve the quality of images by either increasing the resolution using SRGANs or pan sharpening/image fusion.

### 2.1. Generative Adversarial Networks

Since the introduction of GANs by [6], researchers have dedicated a lot of effort to the applications of GANs in different fields with high success rate. The large-scale training of GANs for the generation of high-fidelity natural images is carried out in [8]. The models trained for this study are trained on imagenet at 128 × 128 resolution and are able to achieve a high inception score (IS) and low Frechet inception distance (FID).

The challenging task of image generation from text description is tackled in [9]. The authors decompose the problem into subproblems and construct a stacked GAN architecture that solves the problem in multiple stages. The initial stage produces low-resolution sketch based on the text description, while the second stage uses the sketches and description to generate high-resolution images with photo-realistic details.

Apart from images, GANs have also been applied in the generation of tabular data, as is evident from various GAN approaches focusing on tabular data synthesis [10,11]. A detailed review of different tabular data synthesis approaches based on GANs is presented in [12].

### 2.2. Super-Resolution GANs

Since the introduction of GANs by [6], researchers have dedicated a lot of efforts in the applications of GANs in different fields with a high success rate. The idea of using GANs for improving the quality of images by increasing the resolution of images was introduced by [7]. The ability of SRGANs to recover photorealistic features from highly downsampled images brings the resulting images closer to original high-resolution images than other state of the art approaches. Residual networks are used by [13] for producing super resolution (SR) images. The residual blocks for this approach are used without batch normalization layers to achieve computational edge over other approaches. Similarly, wider residual blocks are used in [14]. This helped improve the accuracy without adding computational overhead.

Low-resolution data are known to significantly hamper the results in image-based tasks. The challenge of dropping model accuracy for face recognition due to low-resolution images is tackled using SRGANs in [15]. The approach replaced the residual blocks in the original SRGAN approach with shuffle blocks to obtain realistic upsampled images with high resolution from low-resolution images.

Video quality of CCTV cameras is signifacntly improved using SRGANs in [16] which reduces the cost of using high-cost CCTV cameras. Another approach for improving the resolution of surveillance footage uses SRGANs [17]. This study sliced the original images into smaller and overlapping images before using SRGANs for upsampling. The resulting high-resolution images are then stitched together to obtain the complete image which is then passed through a contrast equalization filter.

In another study, the use of SRGANs for improving the resolution of images is explored [18]. The results for classification model for vehicle model are compared for low-resolution images and high-resolution images and upsampled using SRGANs. The models for upsampled images outperformed the ones trained for low-resolution images. In medical practices, computerized tomography (CT) images are used for clinical diagnosis, but hardware limitations and time constraints cause CT images in real scenes to have low resolution. These images are upsampled in [19] using SRGANs, which outperformed state-of-the-art baseline methods.

Satellite imagery is upsampled to obtain high-resolution images from low-resolution images using SRGANs in [20]. Attention-based generator is used in this study to enhance the desirable components in uniform regions. The resolution of digital elevation models (DEM) is increased using a GAN-based approach [21]. An adversarially trained discriminator is used to determine whether the image produced by the generator is real or not. All of these SRGANs-based approaches produced highly encouraging results in their respective field of application and learning tasks.

### 2.3. Pan Sharpening

Apart from SRGANs, another popular technique for sharpening low-resolution images is known as pan sharpening, which uses image fusion to generate high-resolution images. Remote sensing images taken from unmanned aerial vehicles, satellites or other such sources are usually low resolution, as the area of interest lies in a small region of the image, e.g., a leaf for disease detection in an image of a larger portion of the field. Remote sensing images discussed in [22] are fused together for obtaining high-resolution images using pan sharpening. Instead of performing fusion in the image domain, this paper uses extracted features to perform fusion and then reconstructs the image from the fused features.

Low-resolution information in multispectral images is fused with high-resolution RGB images in [23]. The resulting images are then analyzed with different vegetation indices for the detection of gramineae weed in rice fields. Another study focusing on the identification of sunflower lodging uses image fusion to obtain high-resolution images with rich information [24]. In this study, low-resolution multispectral images are enhanced using high-resolution visible range images by matching features. Subsequently, rich multispectral images are obtained by fusing these enhanced images with visible range images.

Apart from application in remote sensing and agriculture, image fusion is heavily used for medical imagery in different learning tasks. Pixel level image fusion is performed on medical images using a sparse representation model [25]. Convolutional sparse representations of different components of the images are obtained using a prelearned dictionary, which are then fused with representations of all source images to be reconstructed using the corresponding dictionary.

Convolutional approaches are generally adopted for pan sharpening; however, some approaches using GANs for pan sharpening have also been explored. The use of GANs for pan sharpening in an unsupervised manner is adopted in [26]. The study does not require ground truth and uses spectral and spatial discriminators along with the generator to perform the required image fusion.

While these pan-sharpening approaches have achieved highly accurate quantitative and qualitative results, the need for multispectral bands limits their applicability. GANs-based approaches, however, are readily applicable for different learning tasks. In view of the above, this paper focuses on integrating SRGANs for wheat yellow rust detection to eliminate the need for high-resolution image capturing devices.

## 3. Methodology

This section provides a detailed account of all the image-processing steps along with the deep learning model used to train for wheat yellow rust detection. A complete pipeline of the methodology used in this study is displayed in Figure 1.

### 3.1. Data Acquisition

The data for this study were acquired using smartphone cameras and digital cameras from experimental fields maintained by National Agriculture Research Council (NARC). The fields were grown for different varieties of wheat, which allowed the collection of a wide variety of data. The images were collected during March–April 2021, since that is the time when yellow rust disease in wheat crops appears in Pakistan. A total of 1922 images belonging to three target classes, namely healthy, resistant, and susceptible, were collected using uniform background, which were then annotated under the supervision of domain experts from NARC. The images collected under these conditions have a high resolution, with most images having a size of around 3000 × 2500 pixels on average. A sample image of the raw dataset is shown in Figure 2.

### 3.2. Image Processing

Image processing is an essential part of the wheat yellow rust detection system. A number of different image processing steps are performed to prepare data suitable for training a deep neural network for the learning task.

#### 3.2.1. Image Segmentation

The raw dataset contains a high amount of noise due to different environmental factors such as sunlight, shadows and varying weather conditions. This information proved to be detrimental to the learning task, i.e., wheat yellow rust detection. There was a need for removal of all the additional background noise because the data in its raw form are not suitable for use. The first task in the image processing pipeline was to segment the region of interest, i.e., the leaf portion in the images. This was performed using Otsu’s method [27], which uses maximum likelihood thresholding for segmentation purposes. The resultant image after applying Otsu’s segmentation is shown in Figure 3.

#### 3.2.2. Leaf Cropping

After segmenting the leaf portion of the images using Otsu’s segmentation method, leaf portion in the images was cropped, and the rest of the image was discarded as it did not contain any information critical to the learning task. This was performed by making use of the fact that the pixels that contain the leaf have numerical values in the natural numbers range while the rest of the pixels with no leaf had a value of zero and could be discarded. Images after the cropping of leaf portion were in the most suitable form for further processing, as shown in Figure 4.

#### 3.2.3. Image Downsampling

After image cropping, the images contained the least possible level of noise in them. The next step in the image-processing pipeline was lowering the resolution of raw images. This was achieved owing to the fact that the original resolution of the images was already high, and increasing the resolution further is a very computationally expensive task. Therefore, the images were downsampled to study the effectiveness of SRGANs on classification of wheat yellow rust. The resolution of all the images was reduced manually to avoid adding unnecessary complexity to the methodology and to keep the focus of the study on the feasibility of SRGANs for wheat rust detection. The final resolution of the images is based on the orientation of leaves in the images. As a general rule, the larger axis of the leaf in the image was reduced to 600 pixels, and the pixels of the other axes were reduced in accordance with the original resolution of the image. A sample output image after reduction in resolution is shown in Figure 5.

#### 3.2.4. Image Upsampling

The resolution of these images was then improved using super-resolution GAN (SRGAN). The images were improved by an upsample factor of 4× using the photorealistic SRGAN presented in [7]. The 4× factor was chosen in order to avoid exceeding the capacity of computational resources. A higher upsampling factor should produce better results. The result of SRGAN on a low-resolution image belonging to the resistant class was shown in Figure 6. The difference between the two images (i.e., low resolution and high resolution) was evident to the naked eye, as seen in Figure 5 and Figure 6. This ensured that all the fine details of the image texture were retained in the resultant image and produced a photorealistic image from low-resolution images. The generative adversarial approach following the idea proposed by [6] used a deep generative network with identical residual blocks. The discriminator in the network consisted of eight convolutional layers and was built in a similar fashion to VGG model. The generator and discriminator architectures are shown in Figure 7.

### 3.3. Model Architecture

For the classification of wheat yellow rust into three target classes, i.e., healthy, resistant and susceptible, a three-layered CNN model was used. The selection of a shallow network was due to the limited data availability. The network consisted of two convolutional layers with 20 × 3 × 3 kernels each. A max pooling layer was then used before the classification layer was used to identify the class each image belongs to. Same model architecture was used to train both versions of the data, i.e., low resolution and high resolution. The model architecture was shown in Figure 8.

## 4. Results and Discussion

The training for the model was carried out using training and validation sets. The validation set was created by setting aside 15% of the training data. The model described in Section 3.3 was trained to study the effectiveness of upsampled images using SRGANs for wheat yellow rust classification. The model optimizer for all the models was Adam, and the learning rate was kept at 0.001 to ensure no interference from other factors in the results of the models. The results obtained after training models on both versions of the data are discussed in this section.

### 4.1. Low-Resolution Images

The low-resolution images that were obtained after downsampling the data were used for training the shallow three-layered CNN architecture model. Different simulations were run for a different number of epochs to obtain maximum possible performance on the holdout test set. This number is between 7 and 15 due to the low amount of data. The training for higher epochs generated worse results that did not contribute any useful information to the discussion and therefore were left out of the discussion. The results for these set of simulations were listed in Table 1. The progress of accuracy and loss on training and validation sets for low-resolution images are shown in Figure 9.

The overall performance on test data for all the simulations is similar, but a deeper look into other performance metrics shows that most models are not able to classify resistant class with a high degree of precision. This can be seen in Table 1 which shows the breakdown of models with respect to their performance on resistant class. It is evident from this table that poor performance on resistant class causes the models to have a worse overall performance on test data. The best-performing model (confusion matrix in Figure 10a) produces balanced results in terms of both precision and recall for a resistant class when compared with other models.

### 4.2. High-Resolution Images

Similar to the approach taken when working with low-resolution images, high-resolution images were also trained using different number of epochs to obtain the best possible results. The model for this data was also trained using the same range of epochs as the model for low-resolution images. The results for these trainings are shown in Table 2. The confusion matrix of the best-performing model, trained for eight epochs, is shown in Figure 10b. The accuracy graphs for training accuracy and validation accuracy are shown in Figure 11. The graph shows the training progress for eight epochs, which is the best-performing model on test images.’

The results documented in Table 2 show that the performance of models trained on high-resolution images varies with the best overall accuracy on the test set reaching a maximum of 83%. The reason for the poor performance of other instances of training on this data is the inability of these models to detect the resistant class correctly in most cases and, in some cases, poor performance in detecting healthy class as well. This trend is seen in Table 2, showing precision and recall for resistant class for all the models. These results help us draw strong conclusions about the impact of SRGANs for wheat yellow rust detection which are discussed in the subsequent section.

### 4.3. Impact of SRGANs

The results discussed in Section 4.1 and Section 4.2 show that the models for high-resolution images outperformed the low-resolution image models in terms of best performance in test data. This is evident from the fact that the best overall accuracy for low-resolution models was 75%, while the high-resolution models gave a maximum overall accuracy of 83%. Models trained on high resolution also provide a balanced prediction in terms of precision and recall for all three classes when compared to the models trained on low-resolution images. This is a considerable difference in the performance of the two versions of the data. It shows that SRGANs can provide a significant boost in the performance of deep learning models, especially in the case of low-resolution data.

### 4.4. Comparison of SR Approaches

The approach presented in this paper is also analyzed with other state-of-the-art SR approaches. Deep residual network-based SR approaches, namely enhanced deep residual networks for single-image super-resolution (EDSR) [13] and wide activation for efficient and accurate image super-resolution (WDSR) [14], were used for this analysis. Low-resolution images of the dataset were upsampled, using both the approaches, and separate models were trained using these super-resolution images. The architecture used for these models is the same as the one discussed in Section 3.3. All the hyperparameters were also kept the same for just comparison between the different approaches. The results, both overall and precision, and recall splits for all three classes produced very close results, with the results for SRGAN being marginally better than the other two approaches. The best results on test data obtained from the data obtained using these approaches are listed in Table 3, along with the best results using the approach used in the methodology in the paper. This shows that the application of SRGANs for wheat yellow rust classification is not dependent on the approach used for upsampling images, since it does not produce a significant difference.

## 5. Conclusions and Future Work

This study indicates that super-resolution generative adversarial networks have the potential to aid in the detection of wheat yellow rust disease. This study empirically shows that SRGANs provide a highly capable and effective approach in real-world scenarios, where it is not always possible to have high-resolution images. Digital image processing to remove as much noise as possible is highly advantageous as it ensures the purity of information in the data.

A comparison of the results shows that improving the resolution of images has a direct correlation with the performance of the deep learning models trained for wheat yellow rust detection. The results of low-resolution images max out at 75%, while the results for the high-resolution images, upsampled using SRGANs, improved to 83%. These results can be improved even further if the computing resources have the capacity of handling and upsampling higher-resolution images than the ones used in this study.

## Figures and Tables

**Figure 1 sensors-21-07903-f001:**
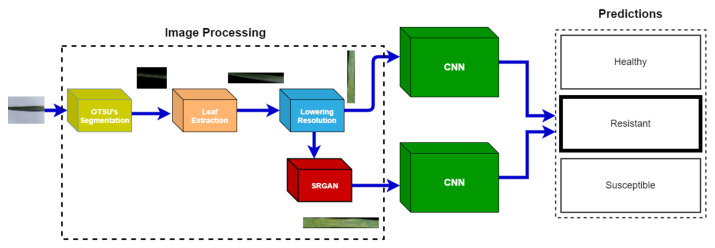
Complete pipeline of the methodology used in this study.

**Figure 2 sensors-21-07903-f002:**
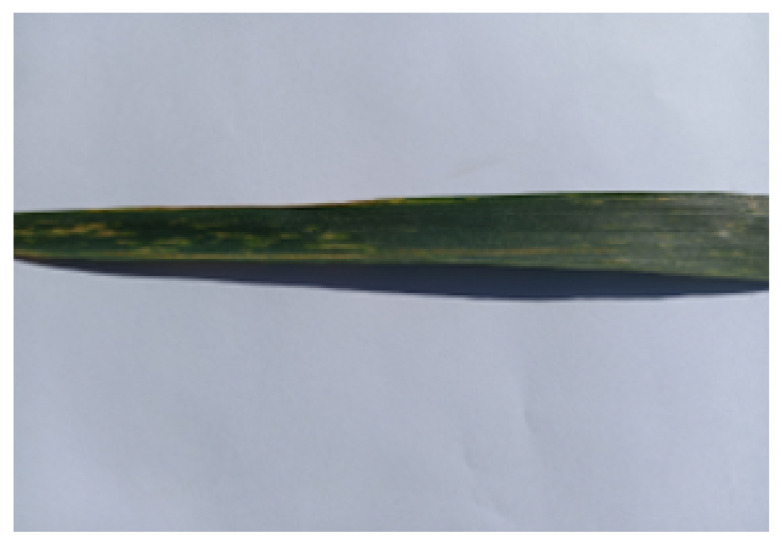
Image sample from the raw dataset.

**Figure 3 sensors-21-07903-f003:**
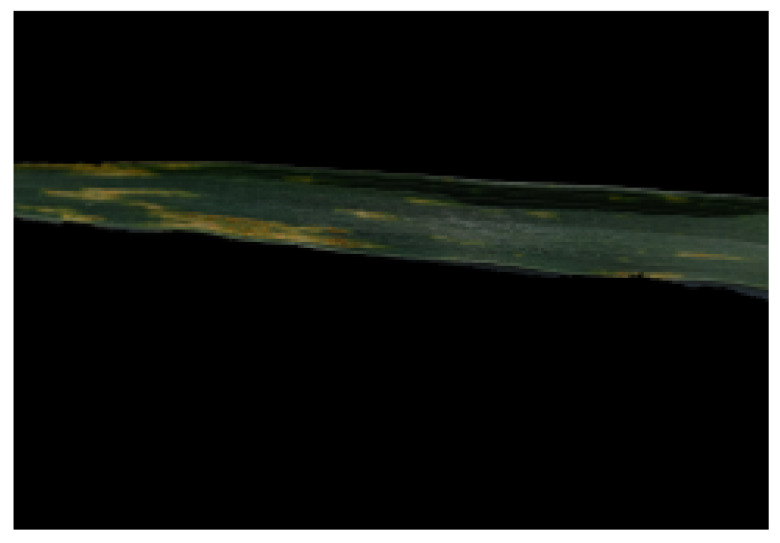
Output of Otsu’s segmentation.

**Figure 4 sensors-21-07903-f004:**
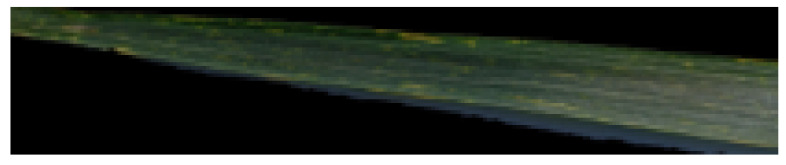
Leaf cropping in the images.

**Figure 5 sensors-21-07903-f005:**
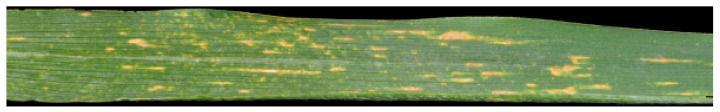
Low-resolution image.

**Figure 6 sensors-21-07903-f006:**
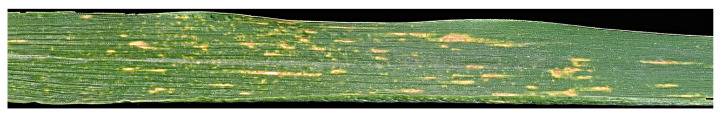
High-resolution image output of SRGAN.

**Figure 7 sensors-21-07903-f007:**
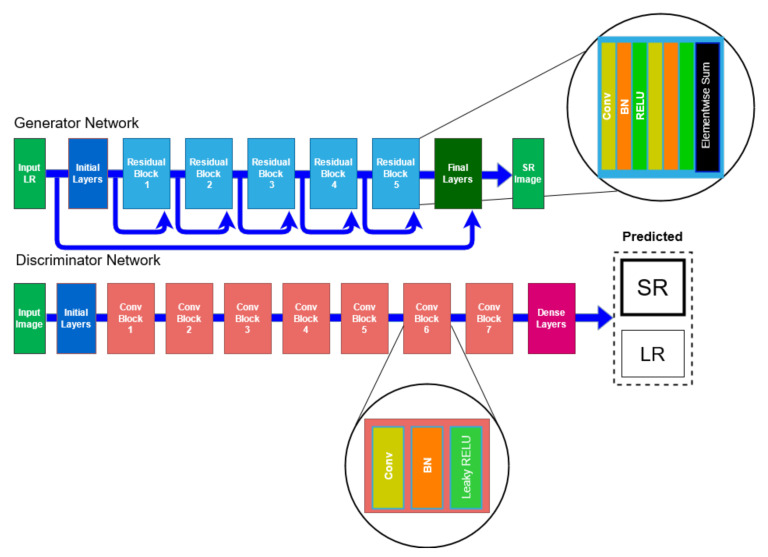
Generator and discriminator architecture of SRGAN inspired by [7].

**Figure 8 sensors-21-07903-f008:**
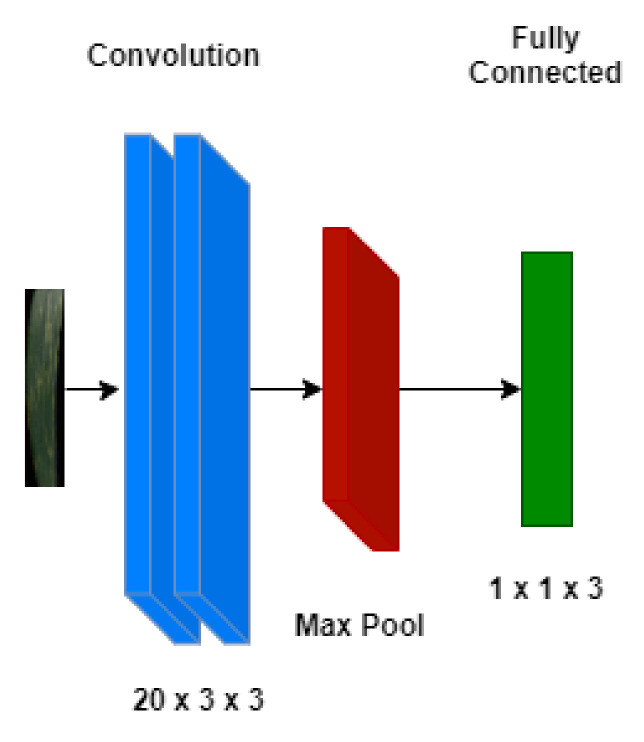
CNN model architecture used in the study.

**Figure 9 sensors-21-07903-f009:**
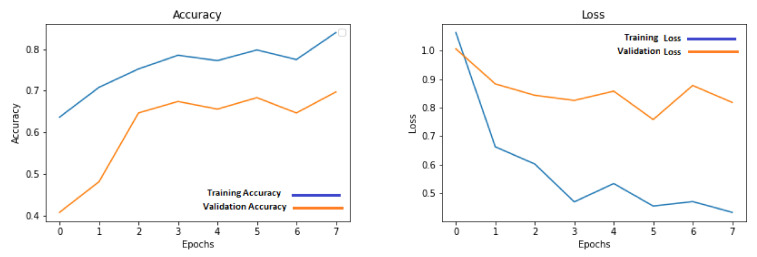
Accuracy and loss graphs for training and validation on low-resolution images.

**Figure 10 sensors-21-07903-f010:**
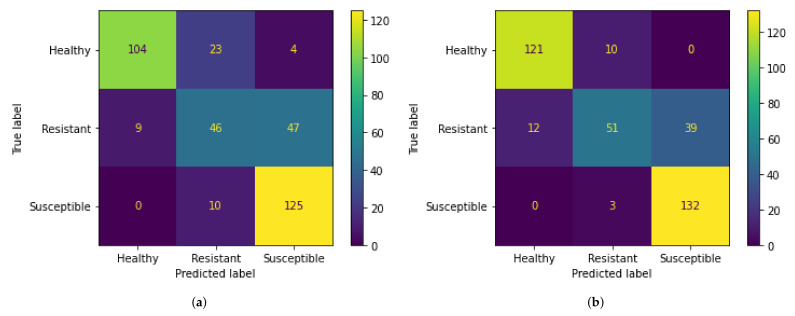
Confusion matrices of the best results for test data. (**a**) Confusion matrix for low-resolution data. (**b**) Confusion matrix for high-resolution data.

**Figure 11 sensors-21-07903-f011:**
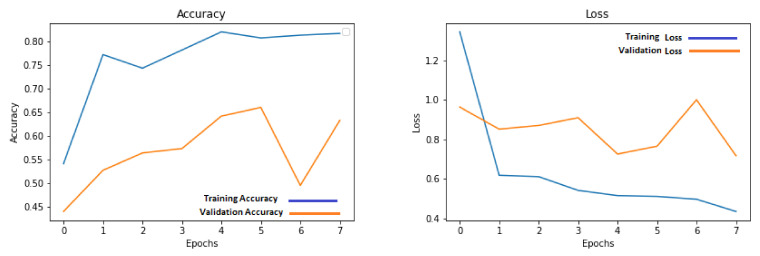
Accuracy and loss graphs for training and validation on high-resolution images.

**Table 1 sensors-21-07903-t001:** Results for models trained on low-resolution images.

Model	Test Accuracy	Healthy		Resistant		Susceptible	
		Precision	Recall	Precision	Recall	Precision	Recall
CNN Epochs = 7	63%	86%	70%	28%	5%	55%	99%
**CNN** **Epochs = 8**	**75%**	**92%**	**79%**	**58%**	**45%**	**71%**	**93%**
CNN Epochs = 9	71%	99%	53%	48%	63%	78%	95%
CNN Epochs = 12	73%	88%	88%	67%	22%	64%	98%
CNN Epochs = 15	71%	78%	82%	54%	33%	71%	88%

**Table 2 sensors-21-07903-t002:** Results for models trained on high-resolution images.

Model	Test Accuracy	Healthy		Resistant		Susceptible	
		Precision	Recall	Precision	Recall	Precision	Recall
CNN Epochs = 7	55%	100%	10%	35%	59%	71%	96%
**CNN** **Epochs = 8**	**83%**	**91%**	**92%**	**80%**	**50%**	**77%**	**98%**
CNN Epochs = 9	80%	84%	86%	65%	58%	85%	90%
CNN Epochs = 12	73%	75%	100%	67%	10%	72%	96%
CNN Epochs = 15	65%	97%	30%	43%	87%	90%	81%

**Table 3 sensors-21-07903-t003:** Best results obtained using different super resolution approaches.

Approach	Test Accuracy	Healthy		Resistant		Susceptible	
		Precision	Recall	Precision	Recall	Precision	Recall
EDSR 5 Epochs	82%	84%	98%	80%	48%	81%	93%
WDSR 9 Epochs	80%	97%	76%	65%	65%	79%	97%
**SRGAN** **8 Epochs**	**83%**	**91%**	**92%**	**80%**	**50%**	**77%**	**98%**

## Data Availability

The data may be requested by reaching out to the authors through email.

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
