# Peer review of "Super Resolution Generative Adversarial Network (SRGANs) for Wheat Stripe Rust Classification"

_sensors, 2021, doi:10.3390/s21237903_

Round 1
Reviewer 1 Report
The author uses Generative Adversarial Network for super-resolution to clarify plant pictures and confirm diseases. It has certain research value in the field of agriculture, but there are still the following problems:
1. The abstract does not need to introduce too much background, but can focus on the algorithm itself.
2. Figure 5 and Figure 7 should be compared together to make the results look clearer.
3. There are too few references. The author should have a comprehensive understanding of the research situation of the research problem.
Reviewer 2 Report
This paper proposes a method for Wheat Stripe Rust Classification based on super resolution methods.
The techniques sound good. Here are some points need to improve this paper.
- All related GANs methods for Super resolution should be added and reviewed.
- The authors should compare the results output of SRGANs with other SR methods by a specific metrics, i.e IS, FID.
Reviewer 3 Report
A good study related to Super Resolution GANs for wheat yellow rust disease detection. Probably, a detailed report about the usage of GAN models at different domain, like generation of realistic photographs (1), text-to-Image translation (2) or even for tabular data generation of cyberattacks (3), would significantly improve the paper reading experience. Attention should be paid in the order of displayed figures, booth in the whole paper and in the text, as well as in some ML terms. Some detailed review notes are given below. Please check the text for spelling mistakes.
Introduction
Line 29. Please add some references for ML and DL approaches that detect wheat yellow rust.
Line 33. “The requirements for different approaches” Please provide some details about the requirements that they are refereed here.
Line 37. “Generative adversarial networks (GAN)” please add a reference here, probably the reference [2] can go here.
The problem that is solved at paper should be mentioned clearer at introduction, for instance what kind of image data will be used?
Methodology
Line 164 “3.2.2. Leaf Cropping” how this is done? A predefined window is used or a detection method? Please provide some more details about this section.
Line 175. Why the downsampling of all the images id done manually? Please provide some more explanation.
Line 182. Why the factor x4 is chosen?
Line 189 The Figure 6 should be mentioned first in the text
Line 192. A figure showing the architecture of the model is needed.
Figure 1. The diagram illustrates that the two CNNs are merged to extract the prediction. However, the two CNNs provide two different predictions, so please update the diagram accordingly.
Results and Discussion
Line 207, 232. “different models, other models”, it is not clear what does this mean. What are the models? There is one model trained at some epoch.
Line 209 “The number of epochs for this set of simulations range from 7 to 15”. Please provide some more description. It is not clear the meaning of this sentence.
Line 221. “training and validation sets” is there also a validation set? Please provide information how the dataset is splitted to training, validation and testing set.
Line 239. “The results discussed in Section 4” -> “The results discussed in subsection 4.1, 4.2”
Figures of confusion matrix, maybe it will be better the two figures to be side by side for a better comparison.
Table 1, 2 of results. Those table can be one after the other for a better comparison.
(1) Brock, A., Donahue, J. and Simonyan, K., 2018. Large scale GAN training for high fidelity natural image synthesis. arXiv preprint arXiv:1809.11096.
(2) Zhang, H., Xu, T., Li, H., Zhang, S., Wang, X., Huang, X. and Metaxas, D.N., 2017. Stackgan: Text to photo-realistic image synthesis with stacked generative adversarial networks. In Proceedings of the IEEE international conference on computer vision (pp. 5907-5915).
(3) Bourou, S., El Saer, A., Velivassaki, T.H., Voulkidis, A. and Zahariadis, T., 2021. A Review of Tabular Data Synthesis Using GANs on an IDS Dataset. Information, 12(9), p.375.
Round 2
Reviewer 2 Report
It's ok for me. I suggest to accept this version.